# Volumetric Imaging of Lung Tissue at Micrometer Resolution: Clinical Applications of Micro-CT for the Diagnosis of Pulmonary Diseases

**DOI:** 10.3390/diagnostics11112075

**Published:** 2021-11-10

**Authors:** Andreana Bompoti, Andreas S. Papazoglou, Dimitrios V. Moysidis, Nikolaos Otountzidis, Efstratios Karagiannidis, Nikolaos Stalikas, Eleftherios Panteris, Vijayakumar Ganesh, Thomas Sanctuary, Christos Arvanitidis, Georgios Sianos, James S. Michaelson, Markus D. Herrmann

**Affiliations:** 1Department of Radiology, Peterborough City Hospital, Northwest Anglia NHS Foundation Trust, Peterborough PE3 9GZ, UK; andreana.bompoti@gmail.com; 2First Department of Cardiology, AHEPA University Hospital, Aristotle University of Thessaloniki, St. Kiriakidi 1, 54636 Thessaloniki, Greece; anpapazoglou@yahoo.com (A.S.P.); dimoysidis@gmail.com (D.V.M.); nickotountzidis@gmail.com (N.O.); stratoskarag@gmail.com (E.K.); nstalik@gmail.com (N.S.); gsianos@auth.gr (G.S.); 3Biomic_AUTh, Center for Interdisciplinary Research and Innovation (CIRI-AUTH), Balkan Center B1.4, 10th km Thessaloniki-Thermi Rd., P.O. Box 8318, GR 57001 Thessaloniki, Greece; eleftherios.panteris@gmail.com; 4Department of Radiology, Medway NHS Foundation Trust, Kent ME7 5NY, UK; vijayakumarganesh@nhs.net; 5Respiratory Department, Medway NHS Foundation Trust, Kent ME7 5NY, UK; thomas.sanctuary@nhs.net; 6Hellenic Centre for Marine Research (HCMR), Institute of Marine Biology, Biotechnology and Aquaculture (IMBBC), 70013 Heraklion, Greece; ceo@lifewatch.eu; 7LifeWatch ERIC, Sector II-II, Plaza de España, 41071 Seville, Spain; 8Department of Pathology, Massachusetts General Hospital, Boston, MA 02114, USA; jamesmichaelsonphd@gmail.com

**Keywords:** micro-CT, pulmonary imaging, lung cancer, chronic obstructive lung diseases, interstitial lung diseases

## Abstract

Micro-computed tomography (micro-CT) is a promising novel medical imaging modality that allows for non-destructive volumetric imaging of surgical tissue specimens at high spatial resolution. The aim of this study is to provide a comprehensive assessment of the clinical applications of micro-CT for the tissue-based diagnosis of lung diseases. This scoping review was conducted in accordance with the PRISMA Extension for Scoping Reviews, aiming to include every clinical study reporting on micro-CT imaging of human lung tissues. A literature search yielded 570 candidate articles, out of which 37 were finally included in the review. Of the selected studies, 9 studies explored via micro-CT imaging the morphology and anatomy of normal human lung tissue; 21 studies investigated microanatomic pulmonary alterations due to obstructive or restrictive lung diseases, such as chronic obstructive pulmonary disease, idiopathic pulmonary fibrosis, and cystic fibrosis; and 7 studies examined the utility of micro-CT imaging in assessing lung cancer lesions (*n* = 4) or in transplantation-related pulmonary alterations (*n* = 3). The selected studies reported that micro-CT could successfully detect several lung diseases providing three-dimensional images of greater detail and resolution than routine optical slide microscopy, and could additionally provide valuable volumetric insight in both restrictive and obstructive lung diseases. In conclusion, micro-CT-based volumetric measurements and qualitative evaluations of pulmonary tissue structures can be utilized for the clinical management of a variety of lung diseases. With micro-CT devices becoming more accessible, the technology has the potential to establish itself as a core diagnostic imaging modality in pathology and to enable integrated histopathologic and radiologic assessment of lung cancer and other lung diseases.

## 1. Introduction

Micro-computed tomography (micro-CT) constitutes a novel non-destructive ex vivo medical imaging modality that enables volumetric imaging of intact surgical lung tissue specimens at high spatial resolution and is thereby capable of providing microscopic insight into alveolar anatomy and pathology [1]. Due to these favourable properties, micro-CT has been used in research settings and helped to advance our understanding of lung histopathology [2]. However, the technology also has significant potential for clinical application [3]. This scoping review provides an overview over the spectrum of clinical micro-CT applications for ex vivo imaging of lung tissue specimens and discusses the impact of the technology on anatomic pathology, the diagnostic workup of surgical lung tissue specimens, and the clinical management of pulmonary diseases.

Similar to CT imaging in radiology, micro-CT imaging relies on an X-ray source and detector assembly in combination with computational image reconstruction algorithms to acquire a series of 2D planar, cross-sectional digital images of the imaging target, which can be computationally processed to generate 3D volumetric models [4]. In contrast to the CT devices used in radiology for in vivo imaging, micro-CT devices used for ex vivo imaging operate at a much smaller scale and with a higher radiation dose, which allows for the acquisition of digital images of lung tissue specimens at micrometre resolution, thereby enabling the diagnosis of pulmonary diseases that manifest itself at the microanatomic scale [5,6]. Within the hospital, micro-CT devices are usually situated in the pathology department, where the surgical tissue specimens are prepared and subsequently imaged with the assistance of specialized medical imaging engineers and technicians [7,8,9,10]. Given its favourable imaging characteristics and rapid image acquisition speed, micro-CT has great potential for clinical applications across a wide spectrum of pulmonary pathologies, including chronic restrictive and obstructive lung disease as well as lung cancer [6,11,12].

However, evidence about the role and utility of micro-CT imaging in the diagnosis and management of lung disease is scarce. To evaluate potential clinical applications of micro-CT imaging of surgical lung tissue specimens, we systematically reviewed the scientific literature and collected all publicly available evidence. Based on our findings, we foresee a pivotal future role of this emerging imaging modality in the interdisciplinary diagnosis of lung disease.

## 2. Materials and Methods

### 2.1. Search Strategy

A scoping review methodology was chosen to achieve this article’s goal to provide insight into the clinical applications of micro-CT imaging for the diagnosis of pulmonary diseases. This was conducted according to the PRISMA Extension for Scoping Reviews (PRISMA-ScR) [13]. Specifically, a systematic literature search was performed in each of the following databases: Scopus, Web of Science, PubMed, ClinicalTrials.gov, and Cochrane CENTRAL until 12 February 2021. Our search included the medical subject heading terms (MeSH): (“micro-CT” OR “X-ray micro-tomography” OR “CT micro-tomography”) AND (“lung*” OR “pulmonary”). Additionally, the PROSPERO database (www.crd.york.ac.uk/prospero) was investigated on 12 February 2021 to identify relevant systematic reviews in order to ensure that there is no other ongoing systematic review on this subject.

### 2.2. Eligibility Criteria

Articles selected by this search were included in this review if (1) they reported on the utilization of micro-CT in pulmonary imaging and if (2) they were published in peer-reviewed scientific journals. An article was excluded if it met at least one of the following criteria: (1) preclinical/animal research, (2) not available in English, and (3) not original research, but review or editorial article type.

### 2.3. Selection of Studies

After importing all articles into the Rayyan reference management program, [14] three reviewers (A.P., D.M. and N.O.) independently screened all titles and abstracts in accordance with the described eligibility criteria to achieve inter-rater reliability. Uncertainty on abstract selection was resolved either by consensus-based discussion or by discussion with a fourth reviewer (A.B.). Potentially eligible full text reports were scrutinized for inclusion by the independent assessment of the reviewers (A.P., D.M. and N.O.), with reasons for exclusion recorded for excluded papers.

### 2.4. Data Extraction and Tabulation

For each eligible article, data were extracted based on (i) country of the author group and year of publication; (ii) study design and population; (iii) objective and main outcome of the study; (iv) type of pulmonary disease investigated; and (v) micro-CT imaging details (cryo-micro-CT scanning method used, manufacturer and model of micro-CT scanner employed, and number of specimens scanned). The tabular and descriptive presentation of our results was based on the type of pulmonary tissue or disease assessed: (i) physiological lung tissue; (ii) chronic obstructive pulmonary disease (COPD); (iii) idiopathic pulmonary fibrosis (IPF); and (iv) others: cancer, congenital anomalies, infections, or lung transplantation-related diseases. Studies reporting micro-CT scanning of lung specimens in more than 1 pulmonary field were inserted in both categories.

### 2.5. Quality Assessment

An assessment of the quality of the included studies was conducted, based on the Newcastle-Ottawa Quality scale [15]. The questions described in this scale were answered independently by three reviewers (A.P., D.M. and N.O.) after categorizing eligible studies into either cohort or cross-sectional studies based on their design. The quality of each study was evaluated in terms of three aspects (selection, comparability, and outcome) with the maximum score of 9 stars. The extracted Newcastle–Ottawa scale score was later converted to three quality levels (good, fair, and poor) in accordance with Agency for Healthcare Research and Quality (AHRQ) standards [16]. In case of discrepancy among the reviewers with respect to the evaluation of the quality of a study, a decision was reached by consensus-based discussion.

## 3. Results

### 3.1. Selection of Eligible Studies

Our initial literature research in the aforementioned databases resulted in 570 potentially eligible articles. The number of studies that remained after each filtering step is depicted in Figure 1. After reading the full-text reports of the potentially eligible studies, a total of 37 studies reporting clinical applications of micro-CT imaging of human lung specimens were included for systematic analysis.

### 3.2. Study Quality

The quality of the included studies was systematically evaluated in accordance with the Newcastle–Ottawa quality scale, as illustrated in Table 1. Quantitative synthesis of the results was not performed, due to the notable heterogeneity of the characteristics of included studies.

### 3.3. Characteristics of Included Studies

The included studies were published between 2005 and 2021. Detailed presentation of the included studies is available in Table 2, Table 3, Table 4 and Table 5; the publications are arranged according to the topic assessed by year of publication. In 9 out of the 37 included studies, micro-CT was used to visualize the microanatomy of the imaged lung specimens independent of a particular disease (Table 2). The majority of studies (*n* = 17) used micro-CT imaging to assess the pathological changes in lung specimens of patients with either COPD (*n* = 11; Table 3) or IPF (*n* = 6; Table 4). The remaining studies used micro-CT imaging in a variety of clinical contexts and for diverse pathologies (Table 5), including lung cancer, alveolar capillary dysplasia, bronchiolitis obliterans syndrome (BOS), restrictive allograft syndrome (RAS), congenital respiratory tract abnormalities, cystic fibrosis, and fungal lung infections.

Nineteen studies were performed in Europe (Belgium, Germany, the Netherlands, Sweden and UK), 13 in North America (USA and Canada), and 5 in Asia (China, Japan and Turkey). The number of the samples tested ranged from 2 to 530 lung specimens (Figure 2). Cryo-micro-CT was reported as a technique of lung tissue imaging in 12 out of the 37 included studies. Different micro-CT scanner models were used in the aforementioned studies, including Bruker Skyscan (*n* = 14), Nikon HMX Metrology (*n* = 8), Explore Locus (*n* = 3), IspeXio SMX (*n* = 3), Med X-Alpha (*n* = 2), and a variety of other models (*n* = 7).

## 4. Discussion

This scoping review presents a comprehensive overview of the current landscape of clinical applications of pulmonary micro-CT imaging (Figure 3). Micro-CT has been used for the quantitative assessment of the pulmonary microanatomy, and in the context of a wide range of lung diseases. The imaging modality has been successfully utilized for high-resolution 3D visualization and morphometric analysis of lung lobules, capillary alveoli, and terminal bronchioles, providing the ability to measure pathological alterations in restrictive and obstructive lung diseases. Determination of positive surgical margins in lung cancer and understanding transplantation-related disorders are two further common clinical applications of micro-CT imaging [1,17,36].

### 4.1. Visualization and Quantitative Analysis of the Lung Microanatomy in Chronic Obstructive and Restrictive Pulmonary Disease

In some application areas, micro-CT has enabled novel insights into the 3D structural alterations in pathological lung specimens. In COPD, for example, micro-CT imaging has made it possible to quantify the loss of terminal bronchioles in early-stage COPD, which has been proposed to pave the way for the emphysematous destruction of the alveoli [30,37]. These alterations are not visible with thoracic multi-detector CT scans used in radiology and understanding these 3D changes at the microscopic level could expand our knowledge of the pathogenesis of pulmonary emphysema [38]. Micro-CT can also be used to acquire images and generate 3D models of the terminal airways and alveoli, allowing the performance of a “virtual bronchoscopy” through an alveolar duct [53]. The produced 3D-imaging data sets acquired ex vivo in pathology could be used in combination with routine high resolution in vivo CT imaging in radiology, facilitating volumetric measurements of tissues from the meter to the micrometre length scale [54]. In the future, the ability of micro-CT to identify pathological terminal and preterminal bronchioles through their anatomical relationship with emerging alveoli in (centrilobular or panlobular) pulmonary emphysema, could make it possible to correlate phenotypic with genetic profiles of patients with COPD and thereby allow a more targeted therapeutic approach [18,34,38].

Furthermore, micro-CT imaging can be used along with micro Optical Coherence Tomography (micro-OCT), CT imaging, and slide microscopy to comprehensively assess pathological changes in IPF [19,55,56]. Micro-CT is capable of detecting fibrosis, even in areas of the lung where CT yields normal findings [39]. Micro-CT-scanned lung samples with minimal pulmonary fibrosis were found to have a significantly lower number of terminal bronchioles when compared to normal control specimens [20,43]. Hence, micro-CT imaging could contribute to our understanding of the pathological process of lung fibrosis, by visualizing structures, such as fibroblast foci, and thereby locate novel therapeutic target sites [42]. Moreover, micro-CT has been utilized to determine disease severity across different lung sites, since fibrosis is known to affect lungs with IPF heterogeneously, which contributes significantly to the investigation of gene expression discrepancies among lung samples at a different stage of fibrosis [32]. Ultimately, a potential future in vivo use of micro-CT in humans could allow earlier diagnosis of fibrotic lung diseases in patients, and could supplement in vivo microscopy imaging modalities such as micro-OCT [39,55,57].

### 4.2. Non-Destructive Imaging of Surgical Specimens for Lung Cancer Diagnosis

Micro-CT allows non-destructive 3D visualization and assessment of lung specimens routinely prepared in pathology, including fresh-frozen tissues as well as formalin-fixed, paraffin-embedded tissue blocks [27,44]. The technology provides 3D spatial tissue context beyond the imaging plane of a physical tissue section specimen evaluated by optical light microscopy and makes it possible to perform micro-structural analysis of relatively large tissue volumes with high diagnostic accuracy, approaching and potentially even surpassing conventional histology [1,34,41].

Micro-CT has also been employed successfully for the study and diagnosis of lung cancer. In particular, the technology facilitates the identification of lymphatic heterogeneity between normal and malignant lung tissues, giving insight into the lymphatic metastatic spread in lung cancer [31]. Additionally, micro-CT can identify alveolar walls in lung cancer areas, with comparable accuracy to conventional histopathologic analysis using optical microscopy of hematoxylin and eosin (H&E) stained tissue section specimens. Therefore, the implementation of micro-CT for ex vivo or in vivo lung imaging could help clinicians circumvent the required preparation of histopathological slides, potentially reducing costs and processing time, as well as the need for diagnostic bronchoscopic or surgical lung biopsies [21]. Micro-CT scanning of fresh surgical lung specimens has already been studied by Troschel et al. and was found to be feasible and potently compatible with the intraoperative setting [28].

Apart from the ability to evaluate surgical margins and lesion size, high-resolution whole-lesion morphologic acquisition could also supplement slide-based histopathology diagnosis, for example by directing the pathologist toward histology sections that are most likely to influence intraoperative decision-making. It should also be noted that conventional histology sectioning could miss significant histopathologic features due to sparse sampling of the specimen. Micro-CT could also expedite the analysis, especially when performed after resection of specimens with metallic fiducial markers placed preoperatively. Optimization of the parameters affecting the perioperative use of micro-CT as well as larger scale studies could lead to the integration of micro-CT into the clinical workflow of lung cancer resection, which could significantly bolster intraoperative decision-making.

### 4.3. Evaluation of Lung Allograft Specimens upon Transplant Rejection or Prior to Transplantation

Interestingly, micro-CT has been also proposed as a tool for the investigation of transplantation-related disease characteristics and for the selection of candidate lung transplants. Firstly, micro-CT can effectively quantify terminal bronchioles and complement histopathology in the identification of pathological alterations in BOS and RAS [45]. Also, in lung contusion and infection cases, micro-CT examination has revealed dense filling of the alveoli [36]. Similarly, Verleden et al. combined the findings from 18F-FDG PET/CT and micro-CT scanning to show that regions with high SUVmax intensity demonstrate end-stage lung fibrosis, whereas low SUVmax corresponds to normal lung tissue without evidence of inflammation or fibrosis [33]. These findings provide deeper insight into the mechanisms underlying lung allograft dysfunction phenotypes. Additionally, in another study by Verleden et al. parenchymal alterations consistent with emphysema could be identified only via micro-CT and not via gross slice or macroscopic CT examination [36]. Therefore, micro-CT could assess and differentiate the cause of rejection in lung transplantation and could be potentially useful in estimating prognosis in redo transplantation candidates. Ultimately, micro-CT could complement the established strict criteria to select suitable donor lungs by measuring the extent of underlying lung injury, which could increase the quality and quantity of lung allografts.

### 4.4. Assessing COVID-19-Related Alterations of Lung Tissue

The emergence of severe acute respiratory syndrome coronavirus 2 (SARS-CoV-2) causing COVID-19 has resulted in hundreds of thousands of casualties worldwide, thereby prioritizing the COVID-19-related research for the rationalization of new treatment strategies. Micro-CT could emerge as another tool in researchers’ arsenal in studying the pathogenesis of the disease and assessing disease severity, as demonstrated in two recent preclinical animal-based studies that investigated the inflammatory response and the extent of lung injury induced by SARS-CoV-2 infection [58,59]. Utilization of micro-CT for ex vivo imaging of infected human lungs could contribute to the expansion of our knowledge of the underlying pathogenesis, the discovery of novel imaging biomarkers, and the development of novel therapeutic approaches for preventing or alleviating lung tissue damage upon SARS-CoV-2 infection.

### 4.5. Example of a Potential Future Micro-CT Application in Lung Cancer Management

Out of the aforementioned applications of micro-CT scanning, its potential involvement in perioperative lung cancer management could be more extensively described. Specifically, besides the perioperative evaluation of tumour size, location, and resection margins within resection specimens, micro-CT could also facilitate identification and quantification of tumour spread through air spaces (STAS), all without compromising specimen integrity. The concept of STAS has first been described in 2008 as a negative prognostic indicator in lung adenocarcinoma [60] and constitutes a post-surgical pathological diagnosis, officially recognized in 2015 as a pattern of invasion by the World Health Organization classification [61]. Due to the poor postoperative outcomes for STAS-positive non-small-cell lung cancer surgeries [62,63], recent research in the field has been orientated towards the association of STAS with preoperative CT features [64,65,66]. Μicro-CT emerges as a potent alternative technology for obtaining volumetric measurements of the tumour and its three-dimensional micro-environment to inform and guide clinical decision making (i.e., type of surgery performed and perioperative management) and thus has the potential to revolutionize thoracic surgical oncology.

## 5. Conclusions

Micro-CT is a promising novel non-destructive imaging technique, which allows the acquisition of volumetric images of lung tissue specimens at high throughput and high spatial resolution without requiring additional specimen preparation. The technology has already been employed in several clinical studies to visualize, measure, and evaluate the characteristics of both normal and diseased lungs at microscopic resolution in 3D. Especially in the context of chronic obstructive and restrictive lung disease, lung cancer, and lung allograft rejection, micro-CT imaging has provided unprecedented insight into the underlying pathological microanatomic changes and has the potential to improve the diagnosis and therapy of patients suffering from these diseases. Given its favourable imaging properties, we anticipate that micro-CT will gain widespread clinical adoption, fill the resolution gap between in vivo CT imaging of the entire lung in radiology and in vitro slide microscopy imaging of lung tissue section specimens in pathology, and ultimately enable 3D pathology across multiple length scales and across disciplinary boundaries.

## Figures and Tables

**Figure 1 diagnostics-11-02075-f001:**
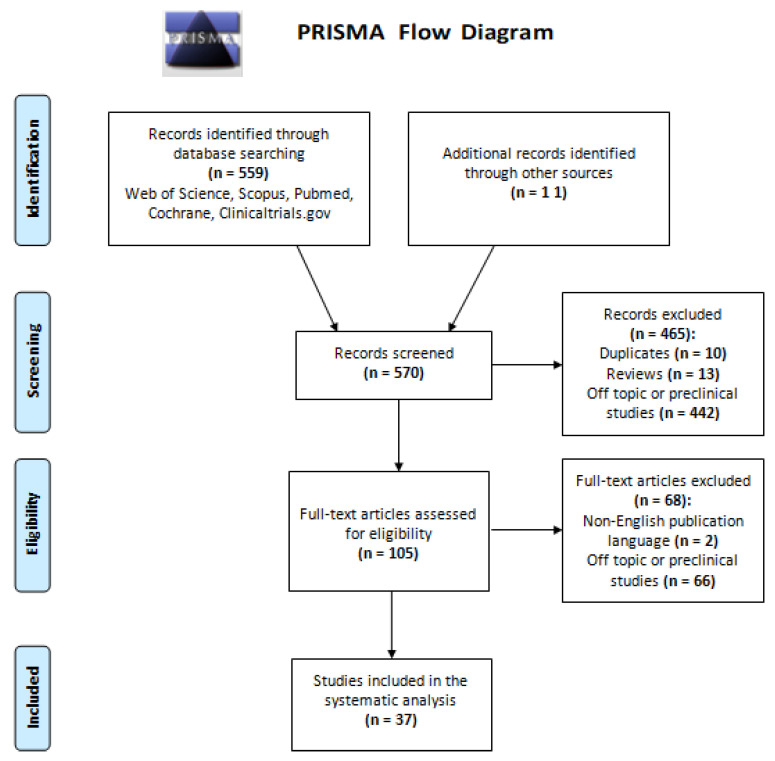
Flowchart of the systematic scoping review in accordance with PRISMA guidelines [13].

**Figure 2 diagnostics-11-02075-f002:**
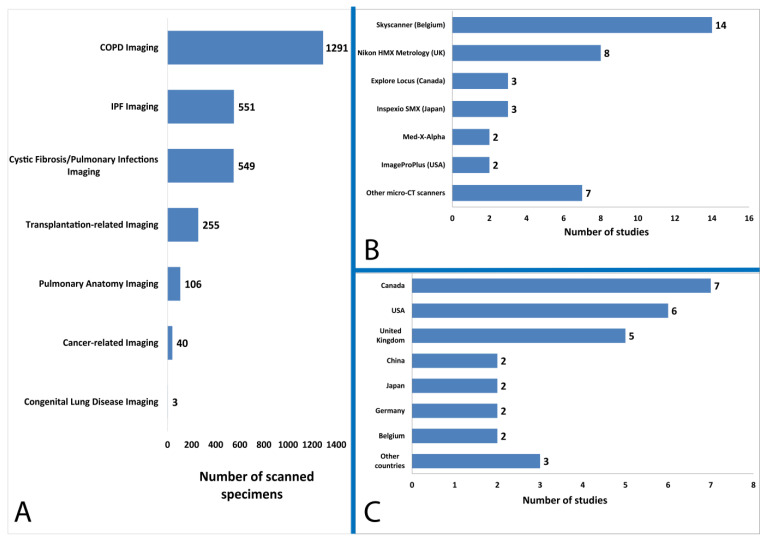
Included study characteristics. (**A**) Number of scanned specimens according to the topic assessed, (**B**) Micro-computed tomography scanners utilized, and (**C**) Geographical distribution of included studies.

**Figure 3 diagnostics-11-02075-f003:**
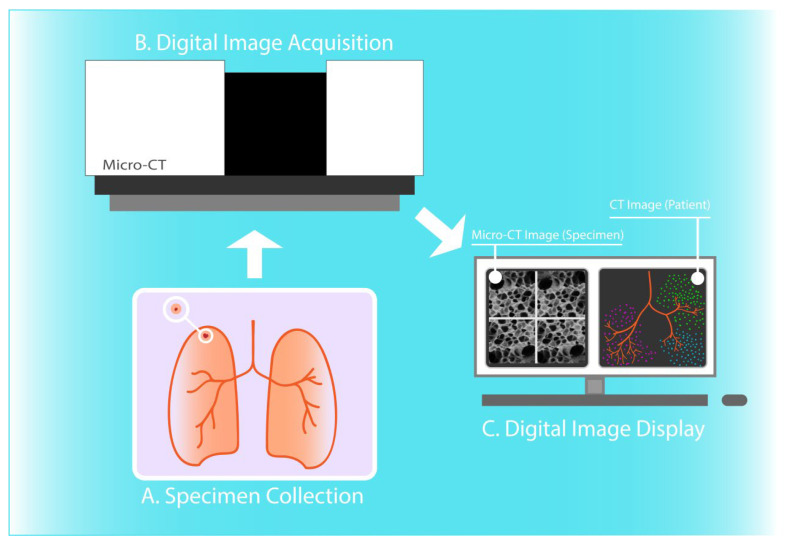
Micro-CT imaging of human lung tissue specimens. Resected lung tissue specimen (**A**) is scanned through micro-computed tomography (micro-CT) scanner (**B**) and 3D images of high spatial resolution are generated after reconstruction (**C**). Herein, 3D micro-CT images of terminal bronchioles and pulmonary acini are displayed (**C left**), in comparison with illustrated CT-derived low attenuation cluster analysis of the airway tree (**C right**).

**Table 1 diagnostics-11-02075-t001:** Newcastle–Ottawa quality assessment scale for observational studies.

Studies	Selection (****)				Comparability (**)		Outcomes (***)		Total Score
	Representativeness of the Sample (*)	Sample Size (*)	Non-Respondents (*)	Ascertainment of the Exposure-Risk Factor (*)	Based on the Design (*)	Based on the Analysis (*)	Assessment of the Outcome (**)	Statistical-Test (*)	
Kayı Cangır, A. [17], 2021	*	*			*		*	*	5
Kirby, M. [18], 2020	*	*		*	*		*		5
Tanabe, N. [19], 2020	*		*		*	*	**		6
Verleden, S. [20], 2020	*	*	*		*	*	*	*	7
Nakamura, S. [21], 2020	*	*		*	*		*	*	6
Verleden, S. [22], 2020	*	*			*		**		5
Norvik, C. [23], 2020		*		*	*		*		5
Umetani, K. [24], 2020	*					*	**		4
Vasilescu, D. [25], 2020		*			*		**		4
Everaerts, S. [26], 2019	*		*	*		*	**	*	7
Katsamenis, O. [27], 2019	*				*		**		4
Troschel, F. [28], 2019	*	*				*	**		5
Shelmerdine, S.C. [29], 2019	*	*			*		*		4
Vasilescu, D. [30], 2019	*		*		*		**		5
Robinson, S.K. [31], 2019		*				*	**	*	5
McDonough, J. [32], 2019	*			*	*		***		6
Verleden, S. [33], 2019	*	*	*		*	*	***		8
Tanabe, N. [34], 2018	*		*		*		**		5
Everaerts, S. [35], 2018		*	*	*		*	**	*	7
Verleden, S. [36], 2017	*		*	*	*		**		6
Suzuki, M. [37], 2017	*	*			*		*	*	5
Tanabe, N. [38], 2017	*	*	*			*	**		6
Mai, C. [39], 2017		*		*		*	**	*	7
Vasilescu, D. [40], 2017	*		*		*		**		5
Guan, C.S. [41], 2016		*				*	*	*	4
Jones, M. [42], 2016	*					*	*	*	4
Boon, M. [43], 2016		*	*	*	*		**		6
Scott, A. [44], 2015		*		*		*	*	*	5
Verleden, S. [45], 2015		*	*	*	*		**	*	7
Zuo, Y.Z. [46], 2013	*		*		*		**		5
Kampschulte, M. [47], 2013	*			*		*	*	*	5
Okubo, Y. [48], 2013	*	*			*		*	*	5
Campell, J. [49], 2012		*	*		*		*	*	5
Litzlbauer, H. [50], 2010		*	*			*	**		5
Hogg, J. [51], 2009	*			*	*		**		5
Hogg, J. [52], 2009	*	*		*	*	*	**	*	8
Watz, H. [53], 2005	*		*		*		*	*	5

The quality of each study was evaluated in terms of three aspects (selection, comparability, and outcome) with the maximum score of 4, 2 and 3 stars (*), respectively.

**Table 2 diagnostics-11-02075-t002:** Studies using micro-CT for visualization and assessment of the pulmonary microanatomy.

Main Author, Year, Country	Study Design	Main Outcome Assessed via Micro-CT	Number of Participants/Specimens (n: 74/105)	Cryo-MicroCT	Micro-CT Scanner
Verleden, S. [22], 2020, Belgium	Cs	Micro-CT was used to assess the number, length, and diameter of terminal bronchioles	32/32	No/not reported	Skyscan 1172 (Bruker, Kontich, Belgium)
Umetani, K. [24], 2020, USA	Cs	Micro-CT was used for whole secondary pulmonary lobule visualization	1/1	No/not reported	BL20B2, SPring-8 (Englewood, CO, USA)
Vasilescu, D. [25], 2020, Canada	Cs	Micro-CT was used as a part of a multi-resolution CT imaging in order to extract specific volumetric findings	13/13	Yes	XT H 225 (Nikon Metrology Inc, Brighton, MI, USA)
Katsamenis, O. [27], 2019, UK	Cs	Micro-CT was used to enable nondestructive 3D-X-ray histology, and to examine its use and benefits in the exemplar of human lung biopsy specimens	2/2	No/not reported	Med-X (Nikon X-Tek Systems Ltd.& Southampton, UK)
Vasilescu, D. [40], 2017, Canada	Cs	Micro-CT was used to image unfixed frozen human lung samples under conditions allowing the tissue to be afterwards used for immunohistochemistry	1/1	No/not reported	Nikon HMX-225 micro-CT scanner (Nikon Metrology, Tring, UK)
Guan, C.S. [41], 2016, China	Cs	Micro-CT was used to retrospectively evaluate short linear shadows connecting pulmonary segmental arteries to oblique fissures in thin-section CT images and determine their anatomical basis	11/11	No/not reported	Siemens micro-CT scanner (Siemens Medical Solutions, Knoxville, TN, USA)
Scott, A. [44], 2015, UK	Cs	Micro-CT was used to visualize, assess and analyze the 3D lung morphology	4/4	No/not reported	Nikon HMX-225 micro-CT scanner (Nikon Metrology, Tring, UK)
Zuo, Y.Z. [46], 2013, China	Cs	Micro-CT was used to describe the normal imaging appearance of pulmonary intersegmental planes compared to thoracic CT scans	10/30	No/not reported	SkyScan 1176 (Bruker, Aartselaar, Belgium)
Litzibauer, H. [50], 2010, Germany	Cs	High-resolution synchrotron-based micro-CT was used to generate a complete dataset of the intact three-dimensional architecture of the human acinus	1/12	No/not reported	X2B beamline, National Synchrotron Light Source (Brookhaven National Laboratories, Germany)

Cs, cross-sectional; micro-CT, micro-computed tomography.

**Table 3 diagnostics-11-02075-t003:** Studies using micro-CT for measurement and evaluation of pathological changes in chronic obstructive pulmonary disease (COPD).

Main Author, Year, Country	Study Design	Main Outcome Assessed via Micro-CT	Number of Participants/Specimens (n: 120/903)	Cryo-MicroCT	Micro-CT Scanner
Kirby, M. [18], 2019, Canada	Cs	Micro-CT was used to estimate the number of terminal bronchioles and their association with total airway count (assessed by multidetector CT)	22/133	Yes	XT H 225 (Nikon Metrology Inc, Brighton, MI, USA)
Everaerts, S. [26], 2019, Belgium	Cs	Micro-CT was used to investigate and compare the airway generations between COPD lungs with and without bronchiectasis and unused donor lungs	21/66	Yes (in 60 specimens)	Skyscan 1172 (Bruker, Kontich, Belgium)
Vasilescu, D. [30], 2019, Canada	Cs	Micro-CT was used to assess terminal bronchioles in emphysema	55/55	Yes	XT H 225 (Nikon Metrology Inc, Brighton, MI, USA)
Tanabe, N. [34], 2018, Canada	Cs	Micro-CT was used to measure the mean linear intercept and the numbers of terminal bronchioles/mL lung in each tissue core	15/15	Yes	XT H 225 (Nikon Metrology Inc, Brighton, MI, USA)
Everaerts, S. [35], 2018, Belgium	Cs	Micro-CT was used to measure surface density and determine the extent of normal tissue within each sample of normal lungs and end-stage COPD lungs.	24/280	Yes	Skyscan 1172 (Bruker, Kontich, Belgium)
Suzuki, M. [37], 2017, USA	Cs	Micro-CT was used to measure the mean linear intercept and the numbers of terminal bronchioles/mL lung in each tissue core	8/61	Yes	eXplore Locus SP MicroCT scanner (GE Healthcare)
Tanabe, N. [38], 2017, Canada	Cs	Micro-CT was used to study small airways pathology in centrilobular and panlobular emphysema and show that these airway alterations are more visible with micro-CT scanners, rather than with thoracic multidetector computed tomography	20/95	No/not reported	Locus SP MicroCT (GE Healthcare, Chicago, IL, USA), Scanco MicroCT35 (Scanco Medical, Brüttisellen, Switzerland), MicroXCT-400 (Zeiss, Oberkochen, Germany), HMX 225ST (Nikon Metrology, Leuven, Belgium)
Kampschulte, M. [47], 2013, Germany	Cs	Micro-CT was used to obtain quantitative volumetric and morphologic information of changes in soft tissue, respiratory tracts and vascularization in fibrotic, emphysematous and non-diseased human lung specimens.	32/32	No/not reported	Not reported
Hogg, J. [52], 2009, Canada	Cs	Micro-CT was used to measure the number and lumen area of terminal bronchioles in COPD lungs	52/530	No/not reported	Micro-CT scanner (Biomedical Imaging Resource, Mayo Clinic, Rochester, MN, USA)
Hogg, J. [51], 2009, Canada	Cs	Micro-CT was used to examine bronchiolar remodelling and alveolar destruction in COPD	8/8	No/not reported	Not reported
Watz, H. [53], 2005, Germany	Cs	Micro-CT was used to investigate the appearance of human lung parenchyma at the structural level of alveoli in a patient with centrilobular emphysema	1/12	No/not reported	CT 20; Scanco Medical, Bassersdorf, Switzerland

Cs, cross-sectional; micro-CT, micro-computed tomography.

**Table 4 diagnostics-11-02075-t004:** Studies using micro-CT for measurement and evaluation of pathological changes in idiopathic pulmonary fibrosis (IPF).

Main Author, Year, Country	Study Design	Main Outcome Assessed via Micro-CT	Number of Participants/Specimens (n: 60/529)	Cryo-MicroCT	Micro-CT Scanner
Tanabe, N. [19], 2020, USA	Cs	Micro-CT was used to examine associations between histopathologic features of usual interstitial pneumonia and IPF in explanted lungs and to measure alveolar surface density, total lung volume taken up by tissue (%), and terminal bronchiolar number	16/96	No/not reported	Skyscan 1172 (Bruker, Kontich, Belgium)
Verleden, S. [20], 2020, Belgium	Cohort	Micro-CT was used to anatomically identify terminal bronchioles and count them per mL of lung tissue	21/240	Yes	Skyscan 1172 (Bruker, Kontich, Belgium)
McDonough, J. [32], 2019, USA	Cs	Micro-CT was used for the assessment of the extent of fibrosis in each sample via measuring alveolar surface density	10/95	Yes	Skyscan 1172 (Bruker, Kontich, Belgium)
Mai, C. [39], 2017, Belgium	Cs	Micro-CT was used to study underlying lung changes responsible for the CT features of IPF and to gain insight into the way IPF proceeds through the lungs and progresses over time	9/94	No/not reported	Skyscan 1172 (Bruker, Kontich, Belgium)
Jones, M. [42] 2016, UK	Cs	Micro-CT was used to characterize fibroblast foci morphology in lung specimens	4/4	No/Not reported	Nikon HMX-225 micro-CT scanner (Nikon Metrology, Tring, UK)
Kampschulte, M. [47], 2013, Germany	Cs	Micro-CT was used to obtain quantitative volumetric and morphologic information of changes in soft tissue, respiratory tracts and vascularization in fibrotic, emphysematous and non-diseased human lung specimens.	22/22	No/not reported	Not reported

Cs, cross-sectional; micro-CT, micro-computed tomography.

**Table 5 diagnostics-11-02075-t005:** Studies using micro-CT for the analysis of pathological lung changes in other disease backgrounds.

Main Author, Year, Country	Study Design	Main Outcome Assessed via Micro-CT	Number of Participants/Specimens (n: 155/763)	Cryo-Micro-CT	Micro-CT Scanner
Kayı Cangır, A. [17], 2021, Turkey	Cs	Micro-CT was used to evaluate pulmonary adenocarcinoma specimens by comparing tumoral and non-tumoral areas and correlating micro-CT findings with hematoxylin and eosin sections.	3/3	No/not reported	Skyscan 1275 (Bruker, Kontich, Belgium)
Nakamura, S. [21], 2020, Japan	Cs	Micro-CT was used to distinguish areas of normal lung tissue and lung adenocarcinoma	10/10	No/not reported	InspeXio SMX-100CT (Shimadzu, Kyoto, Japan)
Norvik, C. [23], 2020, Sweden	Cs	Micro-CT was used to evaluate the micro-anatomy of normal lung tissue and microvascular anomalies of ACD/MPV (alveolar capillary dysplasia with misalignment of pulmonary veins)	2/2	No/not reported	X02DA TOMCAT beamline, Swiss Light Source (Villigen, Switzerland)
Shelmerdine, S.C. [29], 2019, UK	Cs	Micro-CT was used for post-mortem investigation of an excised stenotic infant trachea	1/1	No/not reported	Med-X Alpha (Nikon Metrology, Tring, UK)
Troschel, F. [28], 2019, USA	Cs	Micro-CT was used to peri-operatively evaluate fresh surgical lung resection specimens from patients with a presumptive diagnosis of lung cancer	21/22	No/not reported	Skyscan 1275 (Bruker, Kontich, Belgium)/XT H 225 (Nikon Metrology Inc, Brighton, MI, USA)
Robinson, S.K. [31], 2019, UK	Cs	Micro-CT was used to identify via mathematic modelling and Immunohistochemistry lymphatic heterogeneity within and between lung tissue	2/4	No/not reported	Nikon Metrology micro-CT scanner (Nikon Metrology, Tring Herts, UK)
Verleden, S. [33], 2019, Belgium	Cohort	Micro-CT findings were compared to ^18^F-FDG PET/CT scan findings in a patient with restrictive allograft syndrome undergoing redo transplantation	1/1	Yes	Skyscan 1172 (Bruker, Kontich, Belgium)
Everaerts, S. [35], 2018, Belgium	Cs	Micro-CT was used to measure surface density and determine the extent of normal tissue within each sample of normal lungs and lungs with end-stage cystic fibrosis, chronic hypersensitivity pneumonitis, bronchiolitis obliterans and restrictive allograft syndromes.	46/280	Yes	Skyscan 1172 (Bruker, Kontich, Belgium)
Verleden, S. [36], 2017, Belgium	Cs	Micro-CT was used to assess mass and density of donor lungs, aiding in decision-making to accept or decline lung allografts	28/28	Yes	Skyscan 1172 (Bruker, Kontich, Belgium)
Boon, M. [43], 2016, Netherlands	Cs	Micro-CT was used to quantify the involvement of small and large airways in end-stage cystic fibrosis	18/167	No/not reported	Not reported
Verleden, S. [45], 2015, Belgium	Cs	Micro-CT was used to evaluate lungs from patients with chronic lung allograft dysfunction	24/246	Yes	Skyscan 1172 (Bruker, Kontich, Belgium)
Okubo, Y. [48], 2013, Japan	Cs	Micro-CT was used to evaluate the pathophysiological implications of the reversed CT halo sign in a patient with invasive pulmonary mucormycosis and a patient with invasive pulmonary aspergillosis	2/2	No/not reported	InspeXio SMX-100CT (Shimadzu, Kyoto, Japan)

Cs, cross-sectional; micro-CT, micro-computed tomography.

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
