# Peer review of "Volumetric Imaging of Lung Tissue at Micrometer Resolution: Clinical Applications of Micro-CT for the Diagnosis of Pulmonary Diseases"

_diagnostics, 2021, doi:10.3390/diagnostics11112075_

Round 1
Reviewer 1 Report
The present study has been well designed, the development is fine and it results interesting in this field.
The discussion could be the crucial point, however it should be more detailed. I would add a paragraph regarding an explaition of future prospective and possible application of micro-CT. For example the possible identification of STAS and/or parameters correlated to it.
The manuscript resulted clear and well written.
Reviewer 2 Report
The article reviewed studies that used micro-CT to image human lung tissue. In this study, the authors aimed to assess the clinical applications of micro-CT to diagnose several types of lung diseases based on ex vivo imaging. A total of 570 papers were screened, and 37 of those met their criteria. To select papers, the criteria were 1- it should report pulmonary imaging using micro-CT and 2- be published in peer-reviewed scientific journals. Study criteria were strong enough to exclude studies which might not be translatable for clinical application and only indicated the potential to use micro-CT for clinical purposes. This study provided a good overview of micro-CT's clinical uses to detect lung diseases without destroying samples.
There are minor issues that authors should consider:
1- The abstract needs to be amended to include a statement about whether micro-CT was successful in detecting the lung diseases. A more important question is whether the results of micro-CT were compatible with routine practices or whether they provided more information and details than standard routing diagnosis practices in clinic. This could be added in line 34 before making a conclusion.
2- In figure 1, in the first step, out of 570 studies, authors excluded 442 articles based on the fact that the studies were off topic or preclinical. In the next step, 66 more articles were excluded as off topic or preclinical. Is there a reason these 66 articles weren't excluded earlier along with the 442 articles?
3-Table 1, the table is confusing and it is difficult to understand the study's scoring based on each criterion. It is suggested to organize the table that shows scores for each study across each criterion. The table below is an example that shows clear scores for each study.
studies |
selection |
Comparability Based on the
|
Outcome
|
Total score |
|||||
Representativeness of the sample (*) |
Sample size (*) |
Non-respondents (*) |
Ascertainment of the exposure (risk factor) (*) |
design (*) |
analysis (*) |
Assessment of the outcome (*) |
Statistical test (*) |
||
Kayı Cangır A.[19], 2021 |
* |
* |
0 |
0 |
* |
|
** |
* |
5/9 |
4- statement in line 175 started with “Determination of positive” needs references.
5- In line 202 “micro-OCT” is mentioned for the first time then it should be written as “ micro Optical Coherence Tomography (micro-OCT)”
6- In line 214, the authors claimed “micro-CT could allow earlier diagnosis of fibrotic lung diseases in patients” what evidence the authors have for this statement. When there are no signs of lung disease, however, a biopsy is not commonly performed. Therefore, how can micro-CT be used for earlier diagnosis of fibrotic lung?
7- In section 4.4 authors discussed the potential application of micro-CT to investigate lung injury caused by SARS-CoV-2 by mentioning two recent preclinical animal studies. Since this review focused on the role of Micro-CT in detecting pulmonary diseases for clinical applications, and excluded all preclinical studies, this section was irrelevant to the purpose of the paper.
8- In line 64, there are inappropriate self-citations. References 10 and 11 are not relevant to the statement.
